# The Modified Longitudinal Capsulotomy by Outside-In Approach in Hip Arthroscopy for Femoroplasty and Acetabular Labrum Repair—A Cohort Study

**DOI:** 10.3390/jcm11154548

**Published:** 2022-08-04

**Authors:** Shuang Cong, Jianying Pan, Guangxin Huang, Denghui Xie, Chun Zeng

**Affiliations:** Department of Joint Surgery, Center for Orthopedic Surgery, The Third Affiliated Hospital of Southern Medical University, Guangzhou 510630, China

**Keywords:** femoroacetabular impingement, hip arthroscopy, longitudinal capsulotomy, femoroplasty, labrum repair

## Abstract

Hip arthroscopy is difficult to perform due to the limited arthroscopic view. To solve this problem, the capsulotomy is an important technique. However, the existing capsulotomy approaches were not perfect in the surgical practice. Thus, this study aimed to propose a modified longitudinal capsulotomy by outside-in approach and demonstrate its feasibility and efficacy in arthroscopic femoroplasty and acetabular labrum repair. A retrospective cohort study was performed and twenty-two postoperative patients who underwent hip arthroscopy in our hospital from January 2019 to December 2021 were involved in this study. The patients (14 females and 8 males) had a mean age of 38.26 ± 12.82 years old. All patients were diagnosed cam deformity and labrum tear in the operation and underwent arthroscopic femoroplasty and labrum repair by the modified longitudinal capsulotomy. The mean follow-up time was 10.4 months with a range of 6–12 months. There were no major complications, including infection, neurapraxias, hip instability or revision in any patients. The average mHHS were 74.4 ± 15.2, 78.2 ± 13.7 and 85.7 ± 14.5 in 3 months, 6 months and 12 months after surgery, respectively, which were all better than that before surgery (44.9 ± 8.6) (*p* < 0.05). The average VAS were 2.8 ± 1.2, 1.5 ± 0.6 and 1.2 ± 0.7 in 3 months, 6 months and 12 months after surgery, respectively, which were all lower than that before surgery (5.5 ± 2.0) (*p* < 0.05). The modified longitudinal capsulotomy by outside-in approach is proved to be a safe and feasible method for hip arthroscopy considering to the feasibility, efficacy and security. The arthroscopic femoroplasty and labrum repair can be performed conveniently by this approach and the patient reported outcomes after surgery were better that before surgery in short-term follow-up. This new method is promising and suggested to be widely used clinically.

## 1. Introduction

Femoroacetabular impingement (FAI) is the most common disorder in young adults and patients with high activities, which can lead to inguinal pain and limited motion of hip joint [1]. Clinically, patients with symptom for more than 6 months, failure of conservative therapy and positive finding of labral tear on MRI were considered to be the indications of surgery. The surgical treatment of FAI was femoroplasty and/or acetabuloplasty under hip arthroscopy, to correct the osteophyte of cam deformity and/or pincer deformity, as well as repair the torn labrum [2]. However, only surgeons with high arthroscopic experience could product the hip arthroscopy, due to the hip’s deep location, narrow joint space and high curvature of joint surface, which leading to a very limited arthroscopic view [3].

For the problems above, surgeons proposed many techniques to obtain satisfactory arthroscopic view, in which the most effective technique is the sufficient incision of the hip capsule [4]. In the previous reports, the typical approach is the interportal capsulotomy by the inside-out approach [5]. The hip joint is accessed through portals while the hip in traction and making a capsulotomy inside-out, which making the incision on hip capsule from lateral portal to anterior portal transversely [6].

However, the interportal capsulotomy by inside-out approach has some obvious limitations. Firstly, this approach is complicated to perform, especially in the case of severe pincer deformity. Secondly, the inside-out approach needs the guide of fluoroscopy, leading the patients and surgeons into radiation exposure. Finally, the arthroscopic view obtained by the interportal capsulotomy is not satisfactory enough for consequent femoroplasty and acetabular labrum repair.

Recently, some surgeons proposed a new approach for hip arthroscopy, the so-called outside-in approach [4,7,8,9,10,11]. By this approach, an extracapsular space anterior to the hip joint is established first. Then, surgeons perform the capsulotomy in this extracapsular space and enter the peripheral compartment of hip joint directly without hip traction and fluoroscopy. In the previous studies, many different kinds of capsulotomy were reported, including interportal capsulotomy and T-capsulotomy, as well as longitudinal capsulotomy [6,7].

In our surgical practice, the longitudinal capsulotomy was proved to obtain much better arthroscopic view than either interportal capsulotomy or T-capsulotomy, while it also has some limitations. In the longitudinal capsulotomy, the exposure of the head–neck junction is not enough for femoroplasty, and the acetabular rim is not enough for anchor insertion during the labrum repair. For this reason, we proposed a modified longitudinal capsulotomy which adding a small transverse incision at the proximal capsule to obtain a better arthroscopic view, to make the operating space large enough, and to let the consequent surgical procedure easy to perform.

This study aims to introduce our surgical procedures of the modified longitudinal capsulotomy by outside-in approach, and demonstrate its feasibility and efficacy in arthroscopic femoroplasty and acetabular labrum repair. The hypotheses of this study were the modified longitudinal capsulotomy by outside-in approach can obtain a better arthroscopic view than the traditional surgical approach, and femoroplasty and acetabular labrum repair can be achieved conveniently in the new approach.

## 2. Materials and Methods

### 2.1. Study Design and Participant

With institutional review board (IRB) approval, a retrospective cohort study was performed to collect the patients who underwent hip arthroscopy in our hospital from January 2019 to December 2021. The inclusion and exclusion criteria were as follows:

#### 2.1.1. Inclusion Criteria

① Patients underwent hip arthroscopy in our hospital with diagnosis of FAI and acetabular labrum tear; ② patients underwent femoroplasty and labrum repair during surgery. For all the participants, the diagnosis of FAI and labrum tear were made according to the typical symptoms, physical examination, and radiologic information. Patients with symptom for more than 6 months, failure of conservative therapy, and positive finding of labral tear on MRI were considered to be the indications of the hip arthroscopic surgery.

#### 2.1.2. Exclusion Criteria

① Patients with previous hip surgery history, avascular necrosis, traumatic history around the affected hip and other hip deformities; ② hip osteoarthritis as Tönnis grade > 1 on X-ray image; ③ autoimmune diseases or systemic inflammatory diseases, such as ankylosing spondylitis and rheumatic arthritis.

#### 2.1.3. Sample Size Calculation

A post hoc power analysis was performed using G*Power software (version 3.1; Heinrich Heine University, North Rhine Westphalia, Germany; www.psychologie.hhu.de). Based on the pre-analysis of mHHS before and after the surgery in this study, a minimum of 19 hips were needed to achieve a statistical power of 0.80 by setting α= 0.05 and assuming an effect size of 0.5 to detect significant differences between pre and post operative mHHS.

### 2.2. Preoperative Assessment

For all the patients participating in our study, clinical and radiological assessment was performed in detail. The functional scores for clinical assessment included mHHS (modified Harris hip score) and VAS (Visual analogue score). The mHHS can provide an overall evaluation of the patient-reported clinical function, while the VAS can make a measurement for pain intensity. In the measurement of VAS, a 10-cm-long line is showed with the left end of the scale labeled 0 representing no pain, and the right end labeled 10 representing most severe pain. The patients mark the point on the line based on the severity of the pain they felt, ranging from 0 to 10 [12].

The radiological assessment included X-rays (anteroposterior view, frog view), 3D-CT and MRI (oblique axial view, oblique coronal view, and oblique sagittal view) [13,14,15]. The alpha angle and lateral center edge angle (LCEA) were measured in the X-rays according to previous studies [16,17]. Radiographic parameters were assessed by two surgeons separately blinded to each other (the first and the second authors) with Picture Archiving and Communication Systems (PACS), and the final result was made by a senior surgeon in cases of disagreement. If the alpha angle > 55° or LCEA > 40°, the femoroplasty or acetabuloplasty was performed in surgery [18,19]. The 3D-CT was used to identify the location and size of the cam and pincer deformity. The labrum and cartilage injuries were identified in MRI.

### 2.3. Surgical Procedure

#### 2.3.1. Patient Positioning

All patients underwent hip arthroscopy in the supine position on the orthopedic traction bed, and all patients were under general anesthesia with full muscle relaxation. The perineal post was properly installed. The operative limb was positioned at 15° of internal rotation, neutral flexion/extension and abduction/adduction, while the contralateral limb was positioned at 45° of abduction. The feet were well-padded and fixed in traction boots.

#### 2.3.2. Portal Placement

Before surgery, the outline of the anterior superior iliac spine (ASIS) and the great trochanter were marked before surgery. Three portals were used in all patients. The anterolateral (AL) portal was established between the gluteus minimus and the iliocapsularis muscles, located 1 cm anterior and 1 cm proximal to the great trochanter. The mid-anterior (MA) portal was established 5 cm distal and 1 cm lateral to the ASIS. Finally, the distal anterolateral accessory (DALA) portal was established 5 cm distal and 1 cm anterior to the great trochanter.

#### 2.3.3. Capsulotomy

With the AL portal as an observation portal and MA portal as operation portal, an extracapsular space anterior to the hip joint is established first. After the pericapsular tissue cleared off from the hip capsule, the reflected head of the rectus femoris was located. The longitudinal capsulotomy was performed at the midpoint of the anterior femoral neck, 1 cm lateral and parallel to the rectus femoris using radiofrequency probes. Then, the arthroscopy was entered the hip joint and the femoral head–neck junction was exposed (Figure 1A). The longitudinal incision was extended proximally until the acetabular labrum was exposed (Figure 1B). A 1 cm transverse incision vertical to the longitudinal capsulotomy was performed in the proximal end of longitudinal incision (Figure 1C). A Wissinger rod was used to lift away the pericapsular tissue from DALA portal if necessary.

#### 2.3.4. Acetabuloplasty and Labrum Repair

After the bilateral hip traction was performed, the space of hip joint was increased to approximately 1 cm, so that the arthroscopy can enter the central compartment and the acetabular labrum can expose satisfactorily. After the acetabuloplasty was performed appropriately, if necessary, the acetabular labrum was checked out carefully with probe to identify the torn area. Then 1–2 suture anchors were inserted in the acetabular rim according to the torn size. The torn labrum was fix by mattress suture every 0.5 cm apart.

#### 2.3.5. Femoroplasty

When the management in central compartment completed, the hip traction was released and the arthroscopy entered the peripheral compartment. The perineal post was removed and the operative hip was positioned at 40° of flexion, neutral rotation and adduction/abduction. The modified longitudinal capsulotomy can provide a complete exposure of the femoral head–neck junction and directly identify the location and size of the cam deformity. Then, a 4.0-mm burr was used to demarcate the medial border and extended to the lateral synovial fold (12 o’clock) and the medial synovial fold (6 o’clock). The femoral neck was well visualized and the femoroplasty was performed to provide a smooth transition to the anterior femoral neck. After femoroplasty, impingement tests were performed to check the complete removement of cam deformity. A 45° abduction test is performed in both extension and in 90° of flexion to evaluate possible superolateral impingement. Then, an anterior impingement test is performed by positioning the hip into flexion with maximal internal rotation.

#### 2.3.6. Capsular Closure

At the end of the arthroscopic procedures, the hip capsule was repaired using nonabsorbable, high-tensile strength sutures in a simple side-to-side or shoelace stitches. A total of 2 to 3 stitches were placed to repair the medial and lateral leaflets of the iliofemoral ligament and complete the capsular closure.

### 2.4. Postoperative Rehabilitation

All patients followed the standard protocol of postoperative rehabilitation. Rehabilitation exercises were initiated day 1 postoperatively. Lower extremity resistance exercises were used to begin restoring neuromuscular control and isometric strengthening of the surrounding hip musculature, such as hip abductors and quadriceps. Patients were encouraged to weight-bear as tolerated with crutches after 2 weeks postoperatively. Patients who received labral refixation or/and femoroplasty were ambulated with crutches for 4 weeks and then progressed to full weight-bearing. Range of motion was performed with a continuous passive motion machine, limiting hip rotation and abduction to below 20° and flexion to below 90°.

### 2.5. Data Collection and Statistical Analysis

During the surgical procedures, the capsulotomy time, traction time and overall surgical time were collected. After surgery, each patient underwent the outcome assessment both clinically and radiologically and compared with the preoperative data. The clinical outcomes included mHHS and VAS, as well as postoperative complications in 3 months, 6 months and 12 months’ follow-up. The radiological outcomes included alpha angle and LCEA in X-rays (anteroposterior view, frog view), and the cam deformity was evaluated in 3D-CT.

All data were analyzed using the Stata software (version 13.0; Stata Corp., College Station, TX, USA). Continuous variables were summarized with mean ± standard deviation or median and interquartile range. Continuous variables with normal distribution including alpha angle, LCEA and mHHS were compared using the analysis of variance (ANOVA) and two-sample paired *t*-test. Quantitative data (VAS) which were not normally distributed were compared by the χ^2^ test and two-sample paired Wilcoxon rank-sum test. Statistical significance level was set at *p* < 0.05.

## 3. Results

### 3.1. General Results

Twenty-two postoperative patients who underwent hip arthroscopy in our hospital from January 2019 to December 2021 were involved in this study. The patients (14 females and 8 males) had a mean age of 38.26 ± 12.82 years old. All patients were diagnosed cam deformity and labrum tear in the operation and underwent arthroscopic femoroplasty and labrum repair by the modified longitudinal capsulotomy.

### 3.2. Surgical Results and Radiological Outcomes

The mean capsulotomy time was 12.7 ± 3.5 min, the mean traction time was 36.2 ± 7.2 min and the overall surgical time was 123.6 ± 16.4 min. After surgery, all patients had an alpha angle < 55° and LCEA < 40° in X ray of frog view and anteroposterior view. And the cam deformity was no longer appeared in 3D-CT.

### 3.3. Clinical Outcomes

The mean follow-up time was 10.4 months with a range of 6–12 months. There were no major complications, including infection, neurapraxias, hip instability or revision appeared in any patients. The average mHHS were 74.4 ± 15.2, 78.2 ± 13.7 and 85.7 ± 14.5 in 3 months, 6 months and 12 months after surgery, respectively, which were all better than that before surgery (44.9 ± 8.6) (*p* < 0.05) (Figure 2A). The average VAS were 2.8 ± 1.2, 1.5 ± 0.6 and 1.2 ± 0.7 in 3 months, 6 months and 12 months after surgery, respectively, which were all lower than that before surgery (5.5 ± 2.0) (*p* < 0.05) (Figure 2B). The value of mHHS increased and the value of VAS decreased gradually with the time after surgery.

## 4. Discussion

The modified longitudinal capsulotomy by outside-in approach showed a satisfactory result in this study. Surgeons can complete all of the arthroscopic procedures conveniently after the capsulotomy by this approach. The postoperative mHHS and VAS were both better than before surgery, and patient reported outcomes became better gradually with the time after surgery with no infection, neurapraxias, hip instability or revision appeared in all patients. In spite of the short follow-up time and no control group in this study, the modified longitudinal capsulotomy by outside-in approach was indicated to be a promising method in hip arthroscopy referencing the similar studies previously [6,7]. Moreover, the radiation exposure of surgeons and patients can be avoided because this method did not need intraoperative fluoroscopy.

The outside-in approach was first proposed by Denist et al. in 2005 and proved to be a feasible method in the surgical practice [20]. This approach took the place of puncture approach by seldinger technique in traditional hip arthroscopy. In recent years, studies of hip arthroscopy using the outside-in approach has increased gradually [4,7,8,9,10,11]. This approach can make the operation process quite easy for surgeons performing hip arthroscopy, and all surgical procedures can be achieved without special surgical instruments for hip arthroscopy, or the intraoperative fluoroscopy. The simple procedures make this approach easy to learn and friendly for the beginner of hip arthroscopy. Furthermore, this approach is the best choice for patients with massive pincer deformity, in which the puncture approach can hardly enter the hip joint.

Capsulotomy was a milepost technique in the development of hip arthroscopy, which solved the problems of poor view and difficult procedures in hip arthroscopy [21]. For the different kinds of capsulotomy in previous studies, a systematic review showed 55% performed an interportal capsulotomy while 24% performed a T-capsulotomy [6]. Recently, some surgeons performed the longitudinal capsulotomy and obtained a satisfactory arthroscopic view for the consequent surgical procedures [7,11]. Based on this capsulotomy, we modified the technique by adding a small incision vertical to the longitudinal capsulotomy proximally. This improvement can expose the lesions deep in the hip joint, which help surgeons obtain a good view to observe and a sufficient space to perform the femoroplasty and labrum repair.

The security of capsulotomy is one of the most concerning problems in hip arthroscopy. Therefore, the capsulotomy was conservative and tried carefully when it proposed. After the capsulotomy technique widely used in hip arthroscopy and the development of capsule suture technique, many studies have proved the security of capsulotomy [22,23,24,25,26]. As for the outside-in approach, it avoided the damage of cartilage and labrum caused by puncture in traditional inside-out approach. In the current study, no complication appeared after the hip arthroscopy. On the one hand, the capsulotomy by outside-in approach provided sufficient view and convenient for surgical procedures. On the other hand, the capsulotomy by outside-in approach did not need hip traction and the overall traction time can significantly decrease. Thus, the iatrogenic cartilage or labrum injury and traction-related complications were successfully avoided in this approach.

After the surgical procedures in hip joint, repair of the incised capsule is suggested to avoid postoperative hip instability [22,23,24,25]. By the outside-in approach, an extracapsular space anterior to hip joint is established, which is just convenient for suturing the capsule. Due to the incising direction vertical to iliofemoral ligament, the interportal capsulotomy and T-capsulotomy can injure the iliofemoral ligament and lead to hip instability if the incised capsule not repaired [26,27,28,29]. The modified longitudinal capsulotomy can decrease this injury because the incising direction is paralleled to the iliofemoral ligament. In the present study, the capsular closure was performed in all patients and none of them appeared hip instability after surgery. Therefore, the postoperative rehabilitation processes can be accelerated appropriately after the hip arthroscopy using modified longitudinal capsulotomy.

Considering the feasibility, efficacy and security of the new method, the modified longitudinal capsulotomy by outside-in approach is proved to be a safe and feasible method for hip arthroscopy. This method is easy to perform without special surgical instruments or intraoperative fluoroscopy, and is also quite friendly to beginners of hip arthroscopy. Thus, this new approach is promising and suggested to be widely used clinically.

There were some limitations in this study. Firstly, it is a retrospective cohort study with a relatively small sample size. Secondly, there was no control group of the longitudinal capsulotomy or traditional inside-out approach. Finally, the follow-up time is relatively short. Even though it was found that most patients achieved minimal clinically important difference in 6 months after hip arthroscopy in previous studies [30,31], these limitations above unavoidably decreased the generalizability of the study results. Thus, further study with control group, more sample size and longer follow-up time is needed to check the efficacy of the modified longitudinal capsulotomy.

## 5. Conclusions

The modified longitudinal capsulotomy by outside-in approach is proved to be a safe and feasible method for hip arthroscopy considering to the feasibility, efficacy and security. The arthroscopic femoroplasty and labrum repair can be performed conveniently by this approach and the patient reported outcomes after surgery were better that before surgery in short-term follow-up. This new method is promising and suggested to be widely used clinically.

## Figures and Tables

**Figure 1 jcm-11-04548-f001:**
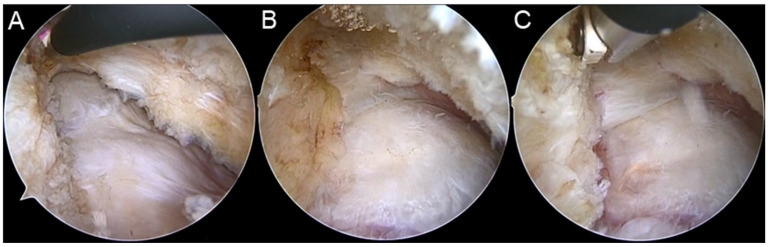
The procedures of modified longitudinal capsulotomy. (**A**) After the longitudinal capsulotomy was performed using radiofrequency probes, the arthroscopy was entered the hip joint and the femoral head–neck junction was exposed. (**B**) The longitudinal incision was extended proximally until the acetabular labrum was exposed. (**C**) A 1 cm transverse incision vertical to the longitudinal capsulotomy was performed in the proximal end of longitudinal incision.

**Figure 2 jcm-11-04548-f002:**
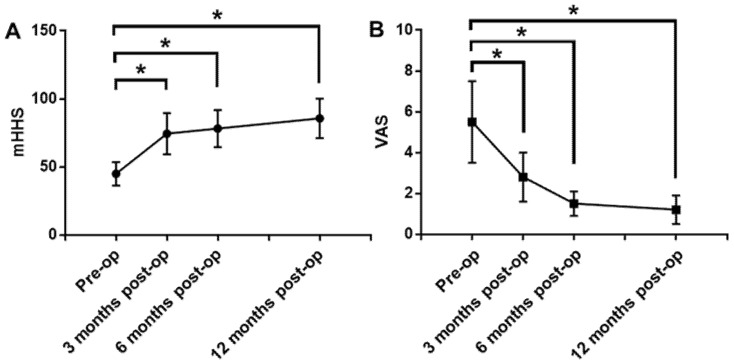
The results of mHHS and VAS. (**A**): The mHHS in 3 months, 6 months and 12 months after surgery were all better than that before surgery (*p* < 0.05). (**B**): The VAS in 3 months, 6 months and 12 months after surgery were all lower than that before surgery (*p* < 0.05). The mHHS increased and the VAS decreased gradually with the time after surgery. *: *p* < 0.05.

## Data Availability

Data are available upon reasonable request from the corresponding author.

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
