# Peer review of "The Modified Longitudinal Capsulotomy by Outside-In Approach in Hip Arthroscopy for Femoroplasty and Acetabular Labrum Repair—A Cohort Study"

_jcm, 2022, doi:10.3390/jcm11154548_

Round 1

Reviewer 1 Report

Reviewer Comments

Thank you very much for the opportunity to review the manuscript submission entitled: he modified longitudinal capsulotomy by outside-in approach in hip arthroscopy for femoroplasty and acetabular labrum repair.

The current paper aims to demonstrate the aims to aimed to propose a modified longitudinal capsulotomy by outside-in approach and demonstrate its feasibility and efficacy in arthroscopic femoroplasty and acetabular labrum repair. The data is interesting, and it has a relevant rationale, however, some limitations and constructive comments are pointed below:

Specific comments

Title and Abstract

·      Include study design in the title

·      Include mean age of the subjects in abstract

·      Include MesH terms as keywords

Introduction

·      The scientific background and rationale for the investigation need to be emphasized. 

·      The hypotheses of the study need to be stated.

Methods

·      Study design should be more appropriate.

·      The inclusion/exclusion criteria should be more detailed and described.

·      How was VAS implemented. Describe in detail.

·      Referencing is required for

o   The radiological assessment included X-rays 98 (anteroposterior view, frog view), 3D-CT and MRI (oblique axial view, oblique coronal 99 views, and oblique sagittal view).

o   The alpha angle and lateral centre edge angle (LCEA) 100 were measured in the X-rays. Radiographic parameters were assessed by two surgeons 101 independently with Picture Archiving and Communication Systems (PACS), and the 102 final results was made by a senior surgeon in cases of disagreement.

Sample size

·      Was sample size calculated?

Discussion

·      Give a cautious overall interpretation of results considering objectives, limitations, the multiplicity of analyses, and results from similar studies.

·      Discuss the generalizability (external validity) of the study results.

Author Response

Replay to Reviewer 1 Comments

TITLE AND ABSTRACT

Include study design in the title

Reply: Thanks for your precious advice, we have added the study design in the title (line 2).

Include mean age of the subjects in abstract

Reply: We have added the mean age of the subjects in abstract (line 25-26).

Include MesH terms as keywords

Reply: We have revised the keywords and the MesH terms were included (line 50-51).

INTRODUCTION

The scientific background and rational for the investigation need to be emphasized.

Reply: We have emphasized the scientific background at line 54-62, and emphasized the rational for the investigation at line 89-92.

The hypotheses of the study need to be stated.

Reply: We have added the hypotheses of the study at line 95-98.

METHODS

Study design should be more appropriate.

Reply: Thanks for your precious advice, this study is a retrospective cohort study with a relatively small sample size and no control group, and the follow-up time is relatively short. In our further study, the control group and longer follow-up time is needed. We have analyzed the limitation of this study in the end of the article (line 327-334).

The inclusion/exclusion criteria should be more detailed and described.

Reply: The inclusion/exclusion criteria were emphasized and described in line 105-122.

How was VAS implemented. Describe in detail.

Reply: We have added the detailed describe of VAS in line 134-137.

Referencing is required for: “The radiological assessment included X-rays (anteroposterior view, frog view), 3D-CT and MRI (oblique axial view, oblique coronal views, and oblique sagittal view). The alpha angle and lateral center edge angle (LCEA) were measured in the X-rays. Radiographic parameters were assessed by two surgeons independently with Picture Archiving and Communication Systems (PACS), and the final results was made by a senior surgeon in cases of disagreement.”

Reply: We have added the referencing at line 138-144.

Sample size: Was sample size calculated?

Reply: The calculation of sample size has been added at line 123-168.

DISCUSSION

Give a cautious overall interpretation of results, considering objectives, limitations, the multiplicity of analyses, and results from similar studies.

Reply: The overall interpretation of results was added at line 267-278.

Discuss the generalizability (external validity) of the study results.

Reply: The generalizability (external validity) of the study results needed to be improved because this study is a retrospective cohort study with a relatively small sample size and no control group, and the follow-up time is relatively short. The discussion of this problem was added at line 330-334.

Reply: Finally, thanks for your precious advices, corresponding revises have been made according to each comment.

Reviewer 2 Report

Dear Authors,

First of all, I would like to congratulate you on the research carried out. However, there are errors that need to be resolved before possible publication in this Journal.

ABSTRACT:
Content related to methodology, results and conclusions should be rewritten to convey objective data and not qualitative and subjective opinions of the authors.

INTRODUCTION:
The authors should adequately contextualize the therapeutic alternatives of FAI.

METHODS:
Statistical analysis should be supplemented by calculating the power of the sample size used.
In addition, the effect sizes of the statistical calculations performed should also be provided.

DISCUSSION:
Again, the authors should expand the text and discuss the clinical impact of the findings of this investigation. In addition, they should contextualize these findings in relation to the postoperative and rehabilitation consequences of these patients.

CONCLUSIONS:
They should also expand on the clinical impact of the findings of this research and their practical significance for the approach to these patients.

GENERAL COMMENTS
They should augment the literature review on the topic and employ relevant bibliographic references on the subject.
Authors should use the journal's template as well as scrupulously follow the journal's editorial and bibliographic reference guidelines.
Authors should take care and correct existing spelling and grammatical mistakes.

Kind regards

Author Response

Replay to Reviewer 2 Comments

ABSTRACT:

Content related to methodology, results and conclusions should be rewritten to convey objective data and not qualitative and subjective opinions of the authors.

Reply: The content related to methodology, results and conclusions in ABSTRACT were rewritten to convey objective data (line 25-47).

INTRODUCTION:

The authors should adequately contextualize the therapeutic alternatives of FAI.

Reply: The therapeutic alternatives of FAI were added in INTRODUCTINO at line 58-62.

METHODS:

Statistical analysis should be supplemented by calculating the power of the sample size used. In addition, the effect sizes of the statistical calculations performed should also be provided.

Reply: The power of the sample size and effect size were added in METHOD at line 123-128.

DISCUSSION:

Again, the authors should expand the text and discuss the clinical impact of the findings of this investigation. In addition, they should contextualize these findings in relation to the postoperative rehabilitation consequences of these patients.

Reply: The clinical impact of the findings was added in DISCUSSION at line 320-325. And we contextualized these findings in relation to the postoperative rehabilitation consequences in DISCUSSION at line 318-320.

CONCLUSIONS:

They should also expand on the clinical impact of the findings of this research and their practical significance for the approach to these patients.

Reply: The clinical impact of the findings and the practical significance were added in CONCLUSION at line 337-343.

GENERAL COMMENTS

They should augment the literature review on the topic and employ relevant bibliographic references on the subject.

Authors should use the journal's template as well as scrupulously follow the journal's editorial and bibliographic reference guidelines.

Authors should take care and correct existing spelling and grammatical mistakes.

Reply: Thanks for your precious advices, corresponding revises have been made according to each comment.

Round 2

Reviewer 2 Report

Dear Authors,

Congratulations on your work, the manuscript has improved considerably and can now be accepted for publication in this Journal.

Kind regards